

# Effects of extracts from various parts of invasive *Solidago* species on the germination and growth of native grassland plant species

Peliyagodage Chathura Dineth Perera[1], Cezary Chmielowiec[1], Tomasz H. Szymura[2] and Magdalena Szymura[1]

[1] Institute of Agroecology and Plant Production, Wrocław University of Environmental and Life Sciences, Wrocław, Poland
[2] Botanical Garden, University of Wrocław, Wrocław, Poland

## ABSTRACT

Allelopathy is an important factor influencing whether an invasive plant species can become successfully established in a new range through disrupting the germination and growth of native plant species. Goldenrods (*Solidago* species) are one of the most widespread invasive taxa in Central Europe of North American origin. Owing to their high environmental impact and wide distribution range, invasive *Solidago* species should be controlled in Europe, and the areas invaded by them should be restored. Numerous studies have reported the allelopathic effects of *Solidago gigantea* and *Solidago canadensis*, but the results are inconsistent regarding differences in the allelopathic effects of particular plant parts and in the sensitivity to *Solidago* allelopathic effects among native species as well as between the two invasive species themselves. In this study, we aimed to analyse the effect of water extracts from *S. canadensis* and *S. gigantea* parts (roots, rhizomes, stems, leaves, and inflorescences) on the germination and initial growth of seedlings of 13 grassland species that typically grow in Central Europe. The tested grassland species differed in susceptibility to *Solidago* allelopathy, with the most resistant species being *Schedonorus pratensis, Lolium perenne, Trifolium pratense, Daucus carota* and *Leucanthemum vulgare*. The inhibitory effect of 10% water extracts from leaves and flowers were stronger than those from rhizomes, roots, and stems without leaves, regardless of the *Solidago* species. Our study results imply that reducing the allelopathic effect of *Solidago* during habitat restoration requires removal of the aboveground parts, including fallen leaves. The allelopathic effects of roots and rhizomes seem to be of secondary importance.

# INTRODUCTION

Allelopathy involves the production of secondary metabolite biochemical substances by one plant that stimulate or inhibit the germination, growth, and development of adjoining or neighbouring organisms (*Rice, 1984*; *Cheema, Farooq & Khaliq, 2013*; *Bachheti et al.,*

Corresponding author
Peliyagodage Chathura Dineth Perera,
chathura.perera@upwr.edu.pl

2020; *Li et al., 2021*). The allelochemicals are present in various plant tissues, primarily inside the cells composing the various plant parts, such as leaves, stems, pollen, flowers, fruits, and roots (*Begum et al., 2019*; *Macías, Mejías & Molinillo, 2019*; *Bachheti et al., 2020*). Allelochemicals can be released to the environment through leaching from leaves and other aboveground plant parts, volatilization, root exudation, and litter decomposition (*Uddin & Robinson, 2017*; *Wang et al., 2021*).

Allelopathy plays a significant role in both natural and agricultural ecosystems by influencing seed germination and the growth of seedlings (*Chon, Kim & Lee, 2003*; *Mushtaq & Siddiqui, 2018*). The inhibition of plant growth caused by allelopathy differs based on the plant tissue (*e.g.*, leaves, stems, roots) from which the allelopathic compounds are released (*Begum et al., 2019*; *Kato-Noguchi & Kato, 2022*). Most research shows that leaf extracts have stronger effects than those from flowers, stems, and roots (*Turk & Tawaha, 2003*; *Siddiqui, Bhardwaj & Meghvanshi, 2009*; *Sodaeizadeh et al., 2009*; *Meiners, 2014*; *Debnath, Debnath & Paul, 2016*; *Mushtaq & Siddiqui, 2018*; *Mangao et al., 2020*), however, it is not absolute and the effect of belowground parts can sometimes be stronger (*Li & Jin, 2010*; *Zivanai, Ronald & Nester, 2019*). The sensitivity to allelochemicals differs considerably among plant species (*Debnath, Debnath & Paul, 2016*; *Sekutowski et al., 2019*; *Mangao et al., 2020*) and even among genotypes within a species (*Meiners, 2014*; *Appiah, Amoatey & Fujii, 2015*). Most allelopathy studies have focused on the interactions between weeds and crops and have described the negative impacts of weeds on crops (*Turk & Tawaha, 2003*; *Mangao et al., 2020*). Allelopathy studies have also focused on aspects of eco-friendly agriculture such as the synthesis of agrochemicals to control pests and diseases, especially in weed management as an alternative to synthetic herbicides (*Chon, Kim & Lee, 2003*; *Macías, Mejías & Molinillo, 2019*; *Bachheti et al., 2020*; *Mangao et al., 2020*; *Li et al., 2021*; *Motmainna et al., 2021*; *Ullah, Khan & Khan, 2021*).

Allelopathy also plays a significant role in the successful establishment and survival of invasive plant species in the ecosystems of new ranges owing to potential for interfering with the seed germination, seedling growth, development, and establishment of native plant species (*Uddin & Robinson, 2017*; *Torawane & Mokat, 2020*). The allopathic effect of some invasive species can be so strong that introduced has been so-called 'novel weapon hypothesis' as an explanation for invasiveness (*Callaway & Ridenour, 2004*). The novel weapons hypothesis explains that the invasion efficiency of an exotic plant species can involve novel biochemical weapons that act as very strong allelopathic agents against resident vegetation. These agents give the invaders an advantage that arises from differences in the coevolutionary histories of plant communities. Allelopathic substances can be relatively ineffective against natural neighbours in the native range of an invader because they are adapted to its presence; however, newly encountered plants in invaded communities lack that adaptation. The exploitation by invaders of the susceptibility of resident species to the allelopathic effect due to evolutionary inadequacy is known as the advantage against resident species hypothesis (*Callaway & Ridenour, 2004*; *Awty-Carroll et al., 2020*). The novel weapons hypothesis explains successful invasions due to allelopathy in cases such as *Centaurea diffusa*, *Centaurea maculosa*, *Mikania micrantha*, and *Alliaria petiolata* in America and China (*Callaway & Ridenour, 2004*; *Chen et al., 2017*) as well

as *Solidago canadensis* in China and Europe (*Chen et al., 2017*; *Wei et al., 2020a*; *Wei et al., 2020b*; *Wei et al., 2020c*) and *Solidago gigantea* in Europe (*Pal et al., 2015*, but see *Del Fabbro, Güsewell & Prati, 2014*).

In this study, we focussed on two invasive species, *Solidago gigantea* Aiton (Giant goldenrod) and *Solidago canadensis* L. (Canadian goldenrod), which are the most widespread invasive species of North American origin in Central Europe (*Meyer, 2022*; *Popay & Parker, 2022*). These *Solidago* species have strong negative environmental impacts due to competition for light, soil nutrients, water, and space, as well as inhibition of native plants through allelopathy (*Ledger et al., 2015*; *Weber & Jacobs, 2005*). Because of clonal growth, *Solidago* species form dense stands and decrease the biodiversity of plants (*Pal et al., 2015*; *Hejda, Pyšek & Jarošík, 2009*), arthropods (*Moroń et al., 2009*; *Lenda et al., 2021*; *Kajzer-Bonk, Szpiłyk & Woyciechowski, 2016*), and birds (*Skórka, Lenda & Tryjanowski, 2010*). The alien *Solidago* are able to invade grasslands, especially recently abandoned ones (*Moroń et al., 2009*; *Fenesi et al., 2015*; *Czarniecka-Wiera, Szymura & Kącki, 2020*; *Szymura, Świerszcz & Szymura, 2022*), strongly influencing their plant species richness and pollinators abundance (*Moroń et al., 2009*; *Fenesi et al., 2015*).

Ample evidence exists regarding allelopathic impact of *Solidago* species on crops and forage grass species. For example, *S. canadensis* extracts can inhibit seed germination and growth performance parameters of *Lactuca sativa* (*Wang, Wu & Jiang, 2019*; *Wei et al., 2020a*; *Wei et al., 2020b*), *Trifolium pratense* (*Zandi et al., 2020*), *Pterocypsela laciniata* (*Wang et al., 2017*), and *Festuca rubra* and *Schedonorus pratensis* (*Karpavičiene, Daniloviene & Vykertaite, 2019*). In addition, *S. gigantea* decreases the germination and growth performance parameters of *Avena sativa*, *Brassica napus* subsp. *oleifera*, and *Helianthus annuus* (*Novak et al., 2018*). Moreover, root extracts from *Solidago* species show inhibitory activity against microorganisms (*Móricz et al., 2020*; *Móricz et al., 2021*). Finally, the allelopathy of *S. canadensis* can reduce the biodiversity of species-rich plant communities and thus increase the susceptibility of a community to further invasion (*Ledger et al., 2015*; *Adomako et al., 2019*).

The content and type of phenolic compounds differ in plant tissues, including leaves, flowers, stem, and roots, and the characteristics can also be variable between *S. canadensis* and *S. gigantea* (*Marksa et al., 2020*; *Kato-Noguchi & Kato, 2022*; *Zhu et al., 2022*). Further, even in the same species, the concentration of allelochemicals can differ based on location (*e.g.*, native *vs.* invasive range). For example, in the case of *S. canadensis*, samples from China had higher allelochemical contents (total phenolics, total flavones, and total saponins) and stronger allelopathic effects than samples from North America (*Yuan et al., 2013*).

Because of the high environmental impact, wide distribution range, and locally high abundance of invasive *Solidago* species, they should be controlled in Europe (*Sheppard, Shaw & Sforza, 2006*; *Fenesi et al., 2015*; *Tokarska-Guzik et al., 2015*). Furthermore, the habitats invaded by them should be restored (*Nagy et al., 2020*; *Szymura, Świerszcz & Szymura, 2022*). A reasonable direction of post-invaded ground restoration is species-rich grasslands (*Szymura, Świerszcz & Szymura, 2022*). Therefore, the selection of grassland species that are resistant to the allelopathic effects of *Solidago* species seems to be important for effective restoration of *Solidago*-invaded lands. The aim of this study was to evaluate the

allelopathic effect of *S. canadensis* and *S. gigantea* plant parts (leaves, stems, inflorescences, roots, and rhizomes) on native grassland species in Central Europe. We hypothesised that (1) the native grassland species differ in terms of seed germination and seedling growth under the allelopathic effects of invasive *Solidago* species, (2) the impacts on germination and growth of native species vary depending on the *Solidago* plant parts used to create water-based extracts, and (3) *S. canadensis* and *S. gigantea* differ in terms of their allelopathic influence on native grassland species. Additionally, we also considered the impact on native species from possible interactions between the three factors (grassland species, extracts of different parts of *Solidago* plants, and *Solidago* species).

## MATERIAL AND METHODS

### Studied species

*Solidago canadensis* and *S. gigantea*, members of the Asteraceae family, are clonal perennial herbs that can form rhizomes. The inflorescences are fasciculate and thyrsoid, the capitula are small and numerous, and the florets are yellow (*McNeil, 1976*). Alien *Solidago* species occur in soils with a wide range of fertility and moisture levels, creating single-species stands or co-occurring with each other (*Weber, 2001*; *Weber & Jacobs, 2005*). The large-range dispersal is realised by numerous, wind-dispersed seeds, whereas short-range dispersal involves vegetative growth through rhizomes (*Weber, 2001*). The species are able to create large stands in abandoned fields and meadows, riparian habitats, forest edges, and unmowed road verges (*Weber, 2001*; *Weber & Jacobs, 2005*; *Fenesi et al., 2015*).

Solidago species contain bioactive compounds such as cytotoxic compounds, phenolic compounds, and flavonoids (*Wandjou et al., 2020*; *Shelepova et al., 2020*; *Kato-Noguchi & Kato, 2022*). Twenty-three phenolic compounds of different phenolic origin were identified in the leaves and inflorescences of *Solidago* species, and *S. gigantea* was found to have higher amounts of the compounds than *S. canadensis* (*Marksa et al., 2020*). Essential oils from *S. canadensis*, composed mainly of mono- and sesquiterpene hydrocarbons, also have phytotoxic potential (*Synowiec et al., 2017*). The highest concentration of phenolic compounds in *S. canadensis* was found in leaves during the blooming stage and in roots during the early growing stage (*Baležentiene, 2015*).

### Plant material

For each *Solidago* species in the current study, we investigated the impact of different plant parts, using aqueous solutions of dried and ground plant material (Fig. S1). The roots, rhizomes, stems, and leaves were collected in July 2020 and the inflorescences in September 2020 in Wrocław, Poland (51°05′57.3″N, 17°04′39.0″E; 51°09′43.7″N, 17°06′54.0″E; and 51°09′40.9″N, 17°06′40.7″E). Thirteen grassland species native to Europe were used in the study (Table 1). These species are typical and widespread in semi-natural grasslands in Central Europe, are important for pollinators, and grow in similar environmental conditions as *Solidago* species. We used species from different plant families, concentrating on species from *Poaceae* and *Fabaceae* families that are the most common in grasslands. The seeds were obtained from Rieger-Hofmann® GmbH company (Blaufelden, Germany) in a ready-to-use form that did not require additional treatments (*e.g.*, freezing) before

**Table 1 Common grassland species used for the experiment.** The nomenclature of plant names according to Euro + Med PlantBase database. The plant species are ordered alphabetically in groups: grasses (Poaceae), legumes (Fabaceae) and other (Apiaceae, Asteraceae, Campanulaceae, Caryophyllaceae).

| No | Species | Abbreviation | Family |
|----|---------|--------------|--------|
| 1 | *Festuca rubra* L. | FR | Poaceae |
| 2 | *Lolium perenne* L. | LP | Poaceae |
| 3 | *Phleum pratense* L. | PhP | Poaceae |
| 4 | *Poa pratensis* L. | PoP | Poaceae |
| 5 | *Schedonorus arundinaceus* (Schreb.) Dumort. | SA | Poaceae |
| 6 | *Schedonorus pratensis* (Huds.) P. Beauv. | SP | Poaceae |
| 7 | *Lotus corniculatus* L. | LC | Fabaceae |
| 8 | *Trifolium pratense* L. | TP | Fabaceae |
| 9 | *Trifolium repens* L. | TR | Fabaceae |
| 10 | *Daucus carota* L. | DC | Apiaceae |
| 11 | *Leucanthemum vulgare* Lam. | LV | Asteraceae |
| 12 | *Campanula patula* L. | CP | Campanulaceae |
| 13 | *Silene flos-cuculi* (L.) Clairv. | SF | Caryophyllaceae |

sowing. The nomenclature of plant names was used according to the *Euro+Med* PlantBase database.

## Allelopathic bioassay

Extracts from the different plant parts of the two *Solidago* species were prepared as 10% aqueous solutions. The concentration was based on previous allelopathic experiments that identified a concentration representative of a high degree of invasion (*Butcko & Jensen, 2002*; *Ravlić, Baličević & Peharda, 2015*; *Novak et al., 2018*; *Sekutowski et al., 2019*). To create the solutions, powdered material (10 g) from each part of the plants was mixed with distilled water (100 ml). The mixtures were set aside in the dark for 24 h at room temperature (20–25 °C) and then filtered, using filter paper to remove plant residues from the solutions (Fig. S2).

For the experiment, Petri dishes (78.54 cm$^2$) with two layers of filter paper were sterilized for 3 h at 120 °C before use. For each grassland species, a sample of 50 seeds was sterilized with 1% NaClO for 15 min, rinsed three times in distilled water, and placed on filter paper to remove excess water. Each sample was then sown onto a Petri dishes and soaked with 8 ml of an extract based on a *Solidago* plant part or with distilled water in the case of the control treatment. Finally, the dishes were closed and placed in a growth chambers (Model—MLR-352H and FRIOCELL, Model—FC 404 EVO; Versatile Environmental Test Chambers SANYO) at 20 °C/10 °C temperature (day/night), with 150 $\mu$mol m$^{-2}$ s$^{-1}$ photosynthetic photon flux density and relative humidity of approximately 70% for 21 days (Fig. S3). During the growth period, the Petri dishes were watered with 1 ml of distilled water every 3 days. Four control treatments for each grassland species and four replications of a particular combination of grassland species (13), *Solidago* species (two), and *Solidago* plant part (five) were prepared (572 Petri dishes in total). For technical reasons, the trials for each species were conducted separately. The experiment was conducted at the Institute
of Agroecology and Plant Production, Wrocław University of Environmental and Life Sciences, Poland from October 2020 to September 2021.

## Measurements

After 21 days, the experiment was terminated, the number of total germinated seeds were calculated for each treatment. Ten random seedlings were selected for each replicate in each trial (40 seedlings per treatment), with fewer seedling selected if germination was poor. Dicotyledonous plant seedling shoot length was measured from the base of the hypocotyl (starting point of the primary root) up to the tip of the longest shoot and monocotyledonous plant seedling shoot length was measured from the point of adventitious root started to the tip of the lengthiest leaf, whereas root lengths of the seedlings in both dicot and monocot were measured up to the ending point of the lengthiest root, using a linear scale and a binocular microscopy, in case of small seedlings. Afterward, the total fresh mass of 10 seedlings from each trial was weighed in grams and calculated the mean value for the single seedling for further analyses.

## Data analysis

The germination percentage of seeds (GP) was determined for each Petri dish separately by using the following formula (*International Seed Testing Association, 1985*):

$$\text{GP \%} = \frac{\text{Number of seeds germinated}}{\text{Total number of seeds plated}} \times 100.$$

The effect of treatments was expressed using 'response index', RI (*Williamson & Richardson, 1988*), which is determined as follows:

If $T > C$ then $RI = 1 - (C/T)$

If $T < C$ then $RI = (T/C) - 1$

where: T—treatment, C—control (average value for four Petri dishes for a species).

RI ranges from $-1$ to $1$, and the negative values indicating inhibition by treatment, while the positive stimulation, relating to control (*Williamson & Richardson, 1988*).

In statistical analysis, to reduce the pseudo-replication problem (*Morrison & Morris, 2000*) we considered a Petri dish to be the smallest, independent sample unit; therefore, results of individual measurements were averaged per dish. In the analyses effect of particular *Solidago* species (*Solidago*), different *Solidago* parts (leaf, flower, root, rhizome, stem—Part), and 13 target grassland species (Species) was considered. Germination inhibition in some cases precluded performing measurements other than an assessment of seed germination, because a lack of seedlings. It resulted in lack of observation in some combination of Species and Part.

Correlations between RI values for particular traits were checked using Spearman's rank correlations coefficient using Past software (*Hammer, Harper & Ryan, 2001*). The non-parametric Scheirer–Ray–Hare test (*Sokal & Rohlf, 1995*) was applied to check significance of differences between groups in RI values. We chose the method because, despite of different data transformations approaches, specific distribution of the data resulting in problems with satisfying assumptions of parametric and semi-parametric methods (normality of residuals and heteroscedastic). The *post hoc*-comparison were done using

Dunn test with Holm adjustment for multiple comparisons. The computation were done in R environment (*R Core Team, 2022*) in rcompanion (*Mangiafico, 2022*) and FSA (*Ogle et al., 2023*) packages for Scheirer–Ray–Hare, and post-hoc tests, respectively.

To clarify the effect of the experiment on a particular plant species, we also performed additional analyses using the RI values only for leaves and flowers of both *Solidago* species together. These analyses were prompted by observed significant differences between the leaves and flowers and the remaining plant parts; the differences were the same for both *Solidago* species (see Results). To better visualise the differences between particular species, we applied discriminant analysis, also considering only the effect of flowers and leaves of both *Solidago* species in Past software (*Hammer, Harper & Ryan, 2001*). *A priori* the discriminatory analysis the RI values were standardized.

# RESULTS

In general, we observed that extracts had a negative effect on seed germination, which was expressed as negative values of response index (RI) (Fig. 1). There were significant correlations between RI values for measured traits, showing that inhibition of germination usually also reduces the biomass and the shoot and root lengths of the studied species (Table 2).

We observed that Part and Species significantly affected all germination and growth characteristics. There was no significant difference between the two *Solidago* species and interactions between treatments. The results of statistical comparisons for the entire experiment are shown in Table 3. The detailed results, with *post hoc* comparisons, are presented in the Appendix (Tables S1–S2). The results consistently show that leaves and flowers cause almost three times stronger inhibition compared with roots, rhizomes, and stems (Fig. 1 and Figs. S4–S7)—for rhizome, root and stem extracts the median of response index was about $-0.25$, while for flower and leaf extracts was around $-0.75$. Considering only the effects of leaf and flower extracts, we observed significant differences between examined species in germination (Fig. 2) as well as RI for other measured traits (Fig. 3). The species *Lolium perenne*, *Schedonorus pratensis*, *Trifolium pratense*, *Daucus carota*, and *Leucanthemum vulgare* were more resistant to the allelopathic effects of *Solidago*, while *Schedonorus arundinacea*, *Festuca rubra*, *Phleum pratense*, *Poa pratensis*, *Lotus corniculatus*, *Trifolium repens*, *Campanula patula*, and *Silene flos-cuculi* were more sensitive (Figs. 2 and 3). In the group of more resistant species the median value of response index was never lower than $-0.50$, while for the sensitive species was always below $-0.75$ threshold. In extremal cases the RI value for *Campanula patula* and *Poa pratensis* was $-0.95$, while for *Trifolium pratense* only $-0.15$. We did not observe an evident differences among particular groups of studied species (grasses, legumes and other) regarding the allelopathic effect. In the group of grasses *Lolium perenne* and *Schedonorus pratensis* seems to be resistant for allelopathic effect, among the legumes *Trifolium pratense* revealed strongest resistance, in the other species group *Daucus carota* and *Leucanthemum vulgare* were most resistant (Figs. 2 and 3, Supplementary Materials).

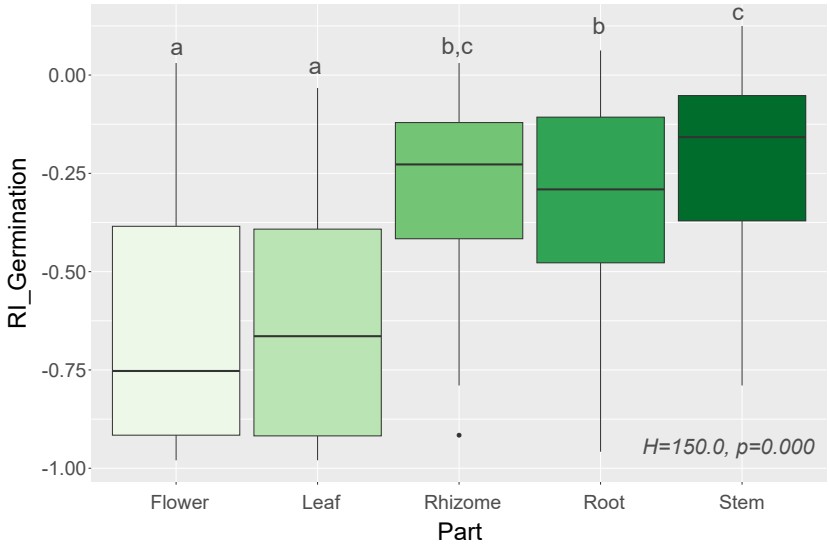

**Figure 1** Response index of grassland species germination (RI_Germination) caused by leaf, flower, root, rhizome, and stem extracts of *Solidago* species (Part), and results of tests (*H* and p). The different letters above boxes indicate significant differ. The different letters above boxes indicate significant differences detected by *post hoc* comparisons.

**Table 2** Results of Spearman's rank correlations (lower part) and associated *p* values (upper part) among response index (RI) values for analysed traits.

|  | RI_Germination | RI_Weight | RI_Shoot | RI_Root |
|---|---|---|---|---|
| RI_Germination |  | 0.000 | 0.000 | 0.000 |
| RI_Weight | 0.65 |  | 0.000 | 0.000 |
| RI_Shoot | 0.42 | 0.50 |  | 0.000 |
| RI_Root | 0.65 | 0.72 | 0.36 |  |

**Notes.**

RI_Germination, response index of germination; RI_Weight, response index of seedling weight; RI_Shoot, response index of soot length; RI_Root, response index of root length.

**Table 3** Results of statistical comparisons (H and p) for response index (RI) for seed germination (RI_germ), seedlings weight (RI_weight), shoot lenght (RI_shoot), root length (RI_root), and their interactions (*p* < 0.05).

|  | df | RI_germ | | RI_weight | | RI_shoot | | RI_root | |
|---|---|---|---|---|---|---|---|---|---|
|  |  | H | p | H | p | H | p | H | p |
| Species | 12 | 210.8 | 0.000 | 94.5 | 0.000 | 228.6 | 0.000 | 146.8 | 0.000 |
| Part | 4 | 150.0 | 0.000 | 211.1 | 0.000 | 87.0 | 0.000 | 210.8 | 0.000 |
| *Solidago* | 1 | 0.3 | 0.577 | 1.2 | 0.279 | 1.9 | 0.172 | 1.2 | 0.272 |
| Species × Part | 46 | 44.9 | 0.517 | 48.5 | 0.261 | 57.3 | 0.071 | 28.8 | 0.952 |
| *Solidago* × Part | 4 | 6.6 | 0.158 | 6.4 | 0.171 | 5.0 | 0.283 | 7.5 | 0.110 |
| *Solidago* × Species | 12 | 9.8 | 0.636 | 11.7 | 0.474 | 5.7 | 0.930 | 23.0 | 0.028 |

**Notes.**

Abbreviations: Species, grassland species, the names are shown in Table 1; Part, extracts of flowers, leaves, stems, rhizomes, and roots parts of *Solidago* plants; *Solidago*, taxon of *Solidago*—*S. gigantea* and *S. canadensis*; df, degree of freedom.

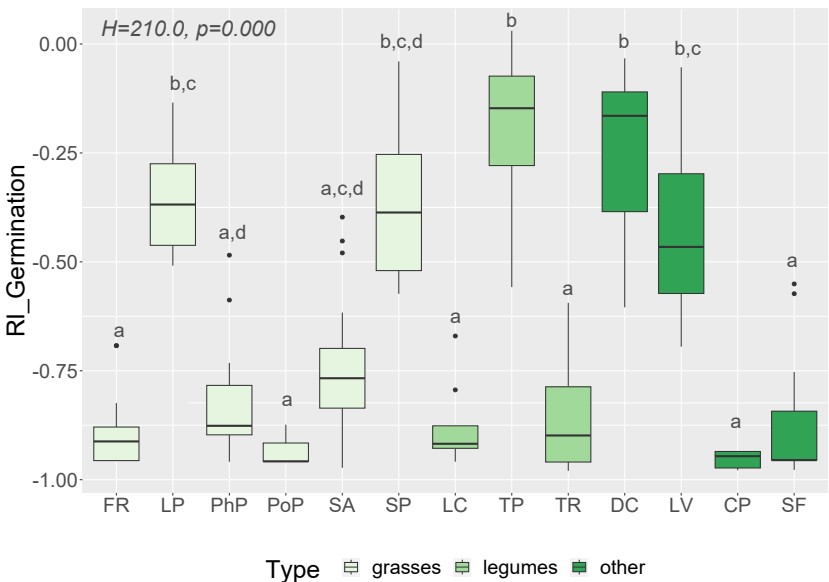

**Figure 2** **Response index of grassland species germination (RI_Germination) caused by *Solidago* allelopathy, and results of tests (H and p).** The different letters above boxes indicate significant differences detected by post hoc comparisons. Species name abbreviations are presented in Table 1. The graph shows the results for combined leaf and flower extracts only.

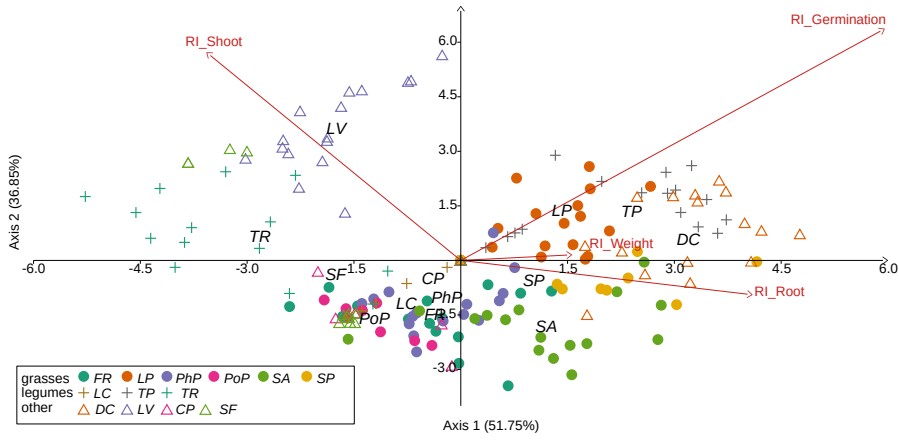

**Figure 3** **Results of discriminant analysis for response index of seed germination, seedling root length, and shoot length and weight of tested grassland species.** Species name abbreviations are presented in Table 1. The graph shows the results for merged leaf and flower extracts only.

In spite of the observed germination inhibition, extracts from root, rhizome, and stem can slightly enhanced the growth of shoots, roots, as well as the biomass for most of the studied species (Figs. S4–S7).

## DISCUSSION

Numerous previous studies report the effect of water extracts from *S. gigantea* and *S. canadensis* on the germination and growth of other plants. Most of the studies focussed on *S. canadensis* (*e.g.*, *Wang, Wu & Jiang, 2019*; *Wei et al., 2020a*; *Wei et al., 2020c*; *Zandi et al., 2020*), while a lower number focussed on *S. gigantea* (*e.g.*, *Pal et al., 2015*; *Baličević, Ravlić & Živković, 2015*; *Sekutowski et al., 2019*). The results are inconsistent because target species, extract concentrations, and tissues from which the extracts were produced varied between studies.

Our results revealed that the tested grassland species differ in susceptibility to *Solidago* allelopathy, which confirms our first hypothesis. The most resistant species in our experiment were *Schedonorus pratensis*, *Lolium perenne*, *Trifolium pratense*, *Daucus carota*, and *Leucanthemum vulgare*. *Chen, Mei & Tang (2005)* and *Megenhardt (2015)* observed that grasses were more sensitive than forbs and legumes to the allelopathic impact of *S. canadensis*. However, in our experiment we did not observe such a pattern, and between *Trifolium repens* and *T. pratense*, legumes belonging to the same genus, *T. pratense* was more resistant than *T. repens* (Figs. 2 and 3). Species detected here as being more resistant to the allelopathic effects of *Solidago* are considered useful in restoring semi-natural grasslands (*Da Silva, Overbeck & Soares, 2017*; *Thiébaut, Tarayre & Rodríguez-Pérez, 2019*; *Zandi et al., 2020*). The obtained results suggest that the above-mentioned species should be prioritised when grasslands are restored on sites invaded by *Solidago*.

Different parts of *Solidago* were assumed to differ in their allelopathic effect (*Marksa et al., 2020*; *Kato-Noguchi & Kato, 2022*; *Zhu et al., 2022*). Allelopathic studies have most often used aboveground parts such as leaves, stems, and flowers, with leaves being most commonly found to have an effect (*Wang, Wu & Jiang, 2019*; *Wei et al., 2020a*; *Wei et al., 2020c*; *Zandi et al., 2020*; *Kato-Noguchi & Kato, 2022*; *Zhu et al., 2022*). Direct comparisons revealed that aboveground parts of *S. canadensis* had a significant allelopathic effect, but the effect of belowground parts was not significant (*Yu et al., 2022*). It was also observed that extracts from *S. canadensis* rhizome stimulated the germination of *Raphanus sativus* seeds and lengthened their shoots (*Anžlovar & Anžlovar, 2012*). In contrast, other studies showed that *S. canadensis* rhizome extract inhibited seed germination and root growth of several native Chinese plant species (*Chen, Mei & Tang, 2005*) and *Zoysia japonica* (*Sun et al., 2022*). In the case of *S. gigantea*, water extracts of its rhizomes and roots increases the dry biomass of *Echinochloa crus-galli* and *Amaranthus retroflexus* (*Sekutowski et al., 2019*), while leaf and stem extracts reduced the growth of *E. crus-galli*. However, *Pal et al. (2015)* showed that root extracts reduced the shoot growth of plant species native to Europe. The observed differences could be due to different reactions of specific target species as well as different concentrations of the extracts. When the extract was at a low concentration (1%), it could even increase the growth of lettuce (*Wang, Wu & Jiang, 2019*; *Wei et al., 2020b*). In addition, *Ye, Meng & Wu (2019)* showed that a water extract of *S. canadensis* shoots up to a concentration of 12.5% could increase the growth of *Zea mays*. However, it should be noted that *Z. mays* seeds are exceptionally large, and the allelopathic effect could weakened due to good isolation of the embryo. In our study, 10% extracts of leaves
and flowers consistently showed stronger effects compared with extracts of rhizomes, roots, and stems without leaves, regardless of the *Solidago* species used. Additional noise in data could be also related to differences in the allelopathic effect of leaves and stems. If stems with leaves are used for preparing solutions (*e.g.*, *Baličević, Ravlić & Živković, 2015*; *Sekutowski et al., 2019*; *Ye, Meng & Wu, 2019*), then the uncontrolled proportion of leaf *versus* stem biomass could change the results obtained. There is great variability in the chemical composition of potentially allelopathic substances between particular parts, seasons and geographical locations in the case of *Solidago* species (See *Marksa et al., 2020*; *Kato-Noguchi & Kato, 2022*; *Zhu et al., 2022*). Therefore, it is impossible to attribute the allelopathic effect observed here to particular chemical substances and differences in their concentrations between plant parts.

The results did not confirm the general hypothesis that allelopathic effects differ between the examined *Solidago* species (Table 2). Previous studies reported inhibition of seed germination and seedling growth in numerous plant species caused by allelopathic effects of both *S. canadensis* (*Wang, Wu & Jiang, 2019*; *Wei et al., 2020a*; *Wei et al., 2020c*; *Zandi et al., 2020*) and *S. gigantea* (*Pal et al., 2015*; *Baličević, Ravlić & Živković, 2015*); however, *Marksa et al. (2020)* found that the leaves of *S. gigantea* contained more active antioxidant compounds than leaves of *S. canadensis*, which suggested that the first species had a stronger allelopathic potential. We assumed that comparing the differences in allelopathic effects of extracts from various parts of *Solidago* and the differences in the susceptibility of target species, the reported previously differences between the two invasive species would have minor practical implications. In practice, the results of a pot experiment also showed that *S. canadensis* and *S. gigantea* have similar competitive abilities (*Szymura & Szymura, 2016*).

If the findings are considered in light of novel weapon hypothesis (*Callaway & Ridenour, 2004*), we can assume that *Solidago* species in invasive range had negative impact on common grassland species, what can facilitate their invasion, and impede the restoration of sites covered by invasive *Solidago*. Nonetheless we did not check the allelopathic effect of the native species to test that the *Solidago* allelopathy should be considered as a novel weapon (*Del Fabbro, Güsewell & Prati, 2014*).

## CONCLUSIONS AND PRACTICAL IMPLICATIONS

Our results confirm that sensitivity to allelopathic effect expressed in germination and seedling growth differ significantly among tested grassland plant species, and it is possible to indicate the species relatively resistant as *Schedonorus pratensis, Lolium perenne, Trifolium pratense, Daucus carota*, and *Leucanthemum vulgare* which should be favoured for site restoration. The results suggest that different *Solidago* plant parts have different allelopathic potential, with strongest influence of flowers and leaves extracts, regardless *Solidago* species studied. Finally we found that there were no substantial differences between *S. canadensis* and *S. gigantea* allelopathic effects, and the difference among plants parts and target grassland species override differentiation between the two *Solidago* species examined. The results yield practical implications for land reclamation. To reduce the allelopathic effect of *Solidago* during habitat restoration, the aboveground parts should be removed, including

fallen leaves. The effects of roots and rhizomes seem to be of secondary importance, and the results of other experiments have shown that restoration is possible without extraction of the belowground parts (*Szymura, Świerszcz & Szymura, 2022*). The difference in allelopathic effects of leaves *versus* stems suggest that these two plant parts should be considered separately, and not mixed, in allelopathic trials. Our results suggest that hard-to-control differences in fractions of leaves and stem biomass in a plant material used to produce extracts may significantly influence experimental outcomes.

## ACKNOWLEDGEMENTS

We would like to thank Dr. Agnieszka Lejman, Dr. Piotr Kuc, and Ms. Zofia Kubińska for their help with the management of the growing chambers. We are also grateful to Professor Zoltán Botta-Dukát and two anonymous reviewers which comments significantly improve the manuscript.

### Funding

This work was supported by the project 'UPWR 2.0:international and interdisciplinary programme of development of Wrocław University of Environmental and Life Sciences', co-financed by the European Social Fund under the Operational Program Knowledge Education Development, contract No. POWR.03.05.00-00-Z062/18 of June 4, 2019 and the Wrocław University of Environmental and Life Sciences (Poland) under the PhD research programme 'Innovative Doctorate' no N070/0008/21. The funders had no role in study design, data collection and analysis, decision to publish, or preparation of the manuscript.

### Grant Disclosures

The following grant information was disclosed by the authors:
UPWR 2.0:international and interdisciplinary programme of development of Wrocław University of Environmental and Life Sciences.
European Social Fund under the Operational Program Knowledge Education Development: POWR.03.05.00-00-Z062/18.
Wrocław University of Environmental and Life Sciences (Poland) under the PhD research programme 'Innovative Doctorate': N070/0008/21.

### Competing Interests

The authors declare there are no competing interests.

### Author Contributions

- Peliyagodage Chathura Dineth Perera conceived and designed the experiments, performed the experiments, analyzed the data, prepared figures and/or tables, authored or reviewed drafts of the article, funding, and approved the final draft.
- Cezary Chmielowiec performed the experiments, authored or reviewed drafts of the article, and approved the final draft.

- Tomasz H. Szymura analyzed the data, prepared figures and/or tables, authored or reviewed drafts of the article, and approved the final draft.
- Magdalena Szymura conceived and designed the experiments, prepared figures and/or tables, authored or reviewed drafts of the article, funding, and approved the final draft.

## Data Availability

The raw measurements are available in the Supplementary File.

## Supplemental Information

Supplemental information for this article can be found online at http://dx.doi.org/10.7717/peerj.15676#supplemental-information.

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
