# Peer review of "Effects of extracts from various parts of invasive Solidago species on the germination and growth of native grassland plant species"

_PeerJ, doi:10.7717/peerj.15676_

## Round 0.1 · original submission · Major Revisions

Dear Dr. Perera,

Three independent experts have made a very thorough analysis of your work. They all agreed that this work could be published in PeerJ, but it had to be substantially revised beforehand. Please read the detailed comments of the reviewers and correct the work accordingly. Please also respond to all comments in the cover letter.

With best regards,

Reviewer 1 ·

Basic reporting

This paper deals with a very interesting and topical problem, the allelopathic effect of invasive Solidago species on grassland plants. The quality of the paper is high. Without going into the strengths of the article, I would like to make a few comments on the weaknesses that have been noted or points that need to be corrected.

Experimental design

What is the nomenclature of plant names used in the text? The source should be indicated, as the taxonomic affiliation of the plants used in the experiment varies quite significantly from one source to another (e.g. circumscription of the genera Lychnis, Festuca).

Validity of the findings

In my opinion, the results section needs to be expanded and clarified. Although most of the most important results are presented in tables and graphs, I missed a coherent and detailed commentary on the results. I would suggest not only to write which species react more strongly to Solidago compounds, but also how much stronger or weaker the reaction is. While differences or similarities can easily be identified from tables, this information cannot always be accurately determined and assessed from graphs.

Additional comments

I would suggest rethinking the conclusions section. The recommendations made are very clear and precise, but I missed the most important conclusions of the study in this chapter. It should be supplemented with real conclusions.

·

Basic reporting

The article gives new valuable data on the allelopathic effect of two widespread invasive species of the Solidago genus. The structure of the paper is clear. The Introduction gives a good summary of the present state-of-art and puts the study in a broader context (e.g., novel weapon hypothesis). The Materials and Methods and Results sections are detailed enough; the results are well-illustrated. The Discussion focuses on the allelopathic effect on Solidago species. A paragraph on outlook to the broader context of invasion theories (e.g., Do these results support or not the novel weapon hypothesis?) could be inserted. The practical implications strengthen the manuscript.

Experimental design

The hypotheses are clearly stated at the end of the Introduction and tested in the subsequent parts of the paper. The research questions are well-defined and fit to the scope of PeerJ. Despite several previous experiments on allelopathy of Solidago species, there are knowledge gaps, partly because of the deficiencies of previous attempts, and partly because Central European species have not been tested.
The experiment is well-designed, and the raw data are reliable. They are made publicly available in the Appendix of the paper. I suggest using CSV (comma-separated values) file format instead of XLSX because, in the long term, the less specific file formats are more robust against changes in software availability.
Unfortunately, the method of statistical analysis is wrong. They first calculated the effect of inhibition (i.e., the relative changes due to allelopathy), and then these values were analyzed by methods assuming the normal distribution of errors. This assumption was not checked (it should be done by drawing diagnostic plots), and I don’t believe this assumption is satisfied because boxplots show skewed distributions.
In my opinion, evaluating relative changes is justifiable for length and weight data but not for germination rate. Therefore, I suggest a log transformation of length and weight values before the analysis. A difference in log scale equals the logarithm of the ratio. Thus, log transformation means focusing on relative instead of absolute changes. When strong germination inhibition preluded performing the measurement, the missing data should not be replaced by an artificial value. I hope that log transformation will improve the normality and homoscedasticity of the residuals. Still, it should be checked, and a more complex model should be fitted where needed. Generalized linear models with binomial distribution and logit link should be fitted for germination data. I would consider including species as a random factor instead of a fixed one because such a model would give a general (i.e., not species-specific) result.

Validity of the findings

Although I did not expect large changes in the meaning of the results due to changes in the analysis suggested above, the content of the Results and the Discussion sections will have to be updated. Therefore, I do not review them in detail, now. In general, their present structure is acceptable.

Additional comments

Finally, some minor comments:
Line 102: Solidago species compete with light, too. I think it may be more important than competition for soil resources.
Line 105: ants is a subgroup of arthropods
line 107: replace “once” by “ones”

Reviewer 3 ·

Basic reporting

Clear and unambiguous, professional English used throughout.
Literature references, sufficient field background/context provided.

The title should state that extracts from various parts of Solidago plants were used.

Experimental design

The methodology requires additions and explanations.
Line 193. Why were the seeds of all species germinated for 21 days? According to the ISTA rules, individual plant species must be germinated under certain temperature and light conditions and within a certain number of days. The germination capacity, for example, of Lolium perenne is determined after 10 days, and of Poa pratensis after 14 days. It seems that the inhibition effect in relation to the individual parameters tested should be assessed at the given times in relation to the control conditions.
Line 193-194. Were all dishes watered with distilled water every 3 days? How much water was used? Were distilled water moistening used, where the seeds germinated under the conditions of using individual extracts (8 ml of the extract was used at the beginning)? If so, the concentration of the extract changed.
Line 201-205. Were 10 seedlings taken from each dish (4 repetitions) of a given germination variant for a given species? How was the length of the hypocotyl (from what point) and the length of the root (from what point and was it the longest root?).
Why was the fresh weight of 10 seedlings determined and not converted to the dry weight of 1 seedling?

Results
The results are very briefly and insufficiently described. There is no comparison between plant species within a given group - e.g. weeds/diots (4 species), Fabaceae (3 species), Poaceae (6 species). Grasses that should be the basis for restoration should be described and ranked according to allelopathic sensitivity/tolerance to the extracts used.
Table 3. No explanation of what "Part" means.
Supplementary
Table 1 - 4. Tables are too large, unstructured data, difficult to analyze. There are large variations between species (they belong to different families) for a given trait (germination, shoot length, root length, seedling weight) and as a result, significance is marked with multiple letters. This should be arranged as suggested above - for each species within a given group. Statistical elaboration should be in relation to selected groups of plants.
Figs. S 4 - S 7. The random arrangement of species in the figures makes it difficult to compare. Species from a given group can be shown in rows, e.g. comparison of Poaceae, Fabaceae, dicots species.

Validity of the findings

No comment

Additional comments

The work requires supplementing and making corrections as suggested and clarifying the issues contained in the review.

---

## Round 0.2 · Minor Revisions

Dear |Dr. Perera,

Your work has been re-evaluated by 2 experts. Both of them stated that the manuscript was considerably improved according to the reviewers’ recommendations. However, one of them is still asking for minor corrections. Special attention was paid by the reviewer to the statistical elaboration. Please read all comments and respond to them in the cover letter.
With best regards,

·

Basic reporting

The authors considerably improved the manuscript following the reviewers’ recommendations.

Experimental design

The new statistical approach is correct, however, its description should be improved on some points. A more condensed and simpler form of response index is RI=(T-C)/max{T, C}.
A more important point is the meaning of C in the formula. I assume that it is the mean of four Petri dishes for each species.
Finally, it should be made more clear that RI, not raw measurements were analysed for all traits.

Validity of the findings

I think the Results and Discussion sections are well-organized, and the results support the conclusions.

Additional comments

Minor comments:
lines 208, 210: “lengthiest leaf” – Do you mean the longest shoot, not the leaf or the leaf at the highest position?
line 257: “we did not find significant effects of Solidago” – It is too technical. I suggest writing that “there was no significant difference between the two Solidago species”, or something similar.
line 292: Leucanthemum, not Leucanhemum

Reviewer 3 ·

Basic reporting

No comment

Experimental design

No comment

Validity of the findings

No comment

Additional comments

The authors made appropriate additions and corrections to the manuscript, including the title. I accept the article in its present form.

---

## Round 0.3 · accepted · Accept

Dear Dr. Perera,

The Reviewer unequivocally stated that all comments were taken into account and the paper can be published in its current version.

Congratulations!

·

Basic reporting

The authors corrected all minor point revealed in the previous review round.

Experimental design

No comment

Validity of the findings

No comment